# Predicting the first steps of evolution in randomly assembled communities

John McEnany[1] & Benjamin H. Good [ID] [2,3,4] ✉

Microbial communities can self-assemble into highly diverse states with predictable statistical properties. However, these initial states can be disrupted by rapid evolution of the resident strains. When a new mutation arises, it competes for resources with its parent strain and with the other species in the community. This interplay between ecology and evolution is difficult to capture with existing community assembly theory. Here, we introduce a mathematical framework for predicting the first steps of evolution in large randomly assembled communities that compete for substitutable resources. We show how the fitness effects of new mutations and the probability that they coexist with their parent depends on the size of the community, the saturation of its niches, and the metabolic overlap between its members. We find that successful mutations are often able to coexist with their parent strains, even in saturated communities with low niche availability. At the same time, these invading mutants often cause extinctions of metabolically distant species. Our results suggest that even small amounts of evolution can produce distinct genetic signatures in natural microbial communities.

Microbes often live in ecologically complex communities containing hundreds of coexisting species[1–4]. As residents of these communities compete with each other, they can evolve over time by acquiring mutations[5–7]. These evolutionary changes can alter the ecological interactions between species, driving shifts in community composition[8–10]. Conversely, the community also creates the context in which organisms evolve, by influencing the structure of the adaptive landscape[11–15]. Understanding how the community influences these evolutionary paths (and vice versa) is a crucial step toward predicting and controlling microbial ecosystems.

Longstanding conceptual models suggest that community structure can impact evolution in different ways. Some models predict that the rate of evolution should decline in larger communities, as more niches are filled by other species[7,16–18]. Others have proposed that diverse communities could create new opportunities for local adaptation, by suppressing key competitors or creating new niches through cross-feeding[11–14,19,20]. These conceptual models also make different predictions about how adaptive mutations will impact their community when they invade. Some mutations will replace their parent strain,

while others can encroach on other species[9,21] or diversify into coexisting lineages[5,6,22]. Each of these behaviors has been observed empirically, yet it remains challenging to predict which should dominate in a given community.

The source of this challenge lies in the niches inhabited by different species, and how mutations alter or move between them. While much progress has been made in small communities where the relevant niches can be explicitly defined[21–24], it is difficult to extend this approach to larger communities like the gut microbiome, where species can compete for many different combinations of resources. In this high-diversity limit, even basic questions about the effects of community structure remain unresolved: how does the benefit of a mutation depend on the diversity of the community and the metabolic overlap between its members? Do mutations primarily compete with their parent strain, or do they continue to stably diversify, as suggested by recent evidence from the gut microbiome[5,6,25] and some in vitro communities[26,27]?

Resource competition models provide a mechanistic framework to address these questions[28–32]. In these models, niches are not defined

[1]Biophysics Program, Stanford University, Stanford, CA, USA. [2]Department of Applied Physics, Stanford University, Stanford, CA, USA. [3]Department of Biology, Stanford University, Stanford, CA, USA. [4]Chan Zuckerberg Biohub – San Francisco, San Francisco, CA, USA. ✉e-mail: bhgood@stanford.edu

in advance but emerge organically through differences in resource consumption[33]. An extensive body of work has used this framework to investigate the process of community assembly, where species compete to colonize a new environment[28,34–37]. These model communities can recapitulate some large-scale features of natural[38–40] and experimental ecosystems[39–42]. By contrast, the evolutionary dynamics that emerge from resource competition are difficult to model with traditional community assembly theory. While some studies have started to explore these effects, previous work has mostly focused on small communities[30,31,43] or the long-term states attained over infinite evolutionary time[30,44]. Both approaches are poorly suited for understanding how a focal species evolves in different community backgrounds, which is the scenario most accessible in experiments. To address this gap, we develop a theoretical framework for predicting the initial steps of evolution in an assembled community with many coexisting species. By extending random matrix approaches from community assembly theory, we derive analytical predictions that describe how the fitness benefits and fates of new mutations scale with the diversity and metabolic overlap of the surrounding community, enabling quantitative tests of the conceptual models above.

## Results

### Modeling first-step mutations in randomly assembled communities

To study the interplay between ecological competition and new mutations in a mathematically tractable setting, we turn to a simple resource competition model[28–32,36,45] where microbes compete for $\mathcal{R} \gg 1$ substitutable resources that are continuously supplied by the environment (Fig. 1a). Each strain $\mu$ in the community is characterized by a resource utilization vector $\mathbf{r}_\mu = (r_{\mu,1}, \ldots, r_{\mu,\mathcal{R}})$, which describes how well it can grow on each of the supplied resources. While our model allows for simple forms of cross-feeding (see below), we neglect additional factors like regulation[46], resource sequestration[47], or antagonistic interactions[48], allowing us to focus on the basic evolutionary pressures imposed by resource competition alone.

In the simplest version of our model, the concentrations of the microbial strains ($\mathbf{n}$) and abiotic resources ($\mathbf{c}$) can be described by the coupled system of equations,

$$\frac{\partial n_\mu}{\partial t} = \sum_{i=1}^{\mathcal{R}} u_i(\mathbf{c}) r_{\mu,i} n_\mu - \delta \cdot n_\mu, \tag{1a}$$

$$\frac{\partial c_i}{\partial t} = K_i - \sum_{\mu=1}^{\mathcal{S}} u_i(\mathbf{c}) r_{\mu,i} n_\mu - \delta \cdot c_i, \tag{1b}$$

where $\delta$ is the dilution rate, $K_i$ is the external supply rate of resource $i$, and $u_i(\mathbf{c}) \cdot r_{\mu,i} \cdot n_\mu$ is the total uptake rate of resource $i$ by species $\mu$ (Supplementary Note 1.1). When the dilution rate is sufficiently low, or biomass sufficiently high, we can coarse-grain over time to obtain an effective model for the relative abundances of the strains ($f_\mu \equiv n_\mu / \sum_\nu n_\nu$),

$$\frac{\partial f_\mu}{\partial t} = f_\mu \left[ \sum_{i=1}^{\mathcal{R}} r_{\mu,i} h_i(\mathbf{f}) - 1 \right], \tag{2}$$

where time is now measured in generations and $h_i(\mathbf{f}) \equiv (K_i / \sum_j K_j) \cdot (\sum_\mu r_{\mu,i} f_\mu)^{-1}$ denotes the local availability of resource $i$ (Supplementary Fig. 1; Supplementary Note 1.2). Simple forms of cross-feeding[39,49] can also be included in this model by replacing the external supply rates $\mathbf{K}$ with an effective value that accounts for internal metabolic conversion (Supplementary Note 1.3).

Previous work has used this model to study the process of community assembly[28,36,37], where $\mathcal{S}$ initial strains arrive in a new environment and form an ecologically stable community containing $\mathcal{S}^* \leq \mathcal{R}$ survivors (Fig. 1b, left). The steady-state values of $\mathcal{S}^*$ and $\mathbf{h}$ depend on

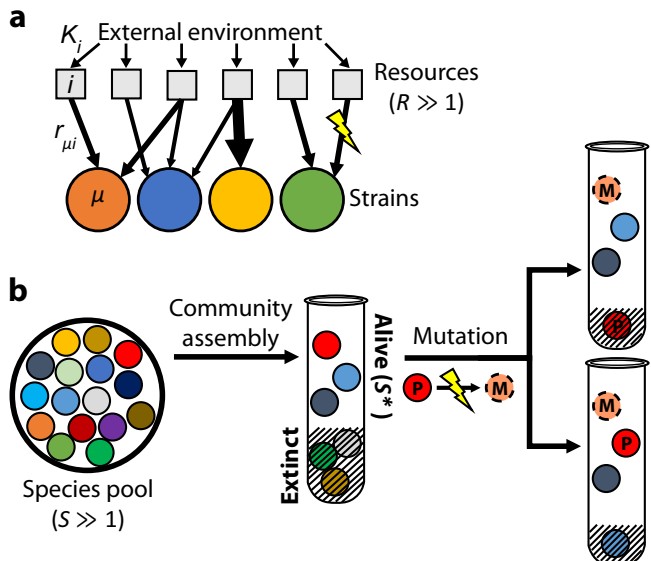

**Fig. 1 | Modeling the first steps of evolution in a randomly assembled community that competes for substitutable resources. a** Microbial strains compete for $\mathcal{R}$ resources that are continuously supplied by the environment at rates $K_i$. Each strain $\mu$ has a characteristic set of uptake rates $r_{\mu,i}$ (arrows), which can be altered by further mutations. **b** A local pool of $\mathcal{S}$ initial species, whose phenotypes are randomly drawn from a common statistical distribution, self-assembles into an ecological equilibrium with $\mathcal{S}^* \leq \mathcal{S}$ surviving species (left). A new mutation ($M$) arises in one of the surviving species ($P$); if the mutation provides a fitness benefit, its descendants can either replace the parent strain (top right) or stably coexist with the parent, potentially driving another species to extinction (bottom right).

the environmental supply rates $\mathbf{K}$ and the resource preferences $\mathbf{r}_\mu$ of the $\mathcal{S}$ initial strains. While it is difficult to measure these kinetic parameters directly, past research has shown that emergent features of large ecosystems can often be mimicked by randomly drawing the uptake rates of each strain from a common statistical distribution[28,29,34–36].

For concreteness, we will initially focus on the binary resource usage model from ref. 36, in which each strain has an independent probability $\mathcal{R}_0/\mathcal{R}$ of utilizing each resource, and an overall uptake budget $X_\mu \equiv \log \sum_i r_{\mu,i}$ that is independently drawn from a Gaussian distribution (Supplementary Note 1.4). However, our results will apply for a variety of different sampling schemes, which we verify in Supplementary Figs. 3, 8 and 9, and Supplementary Note 4.

Individual realizations of this model produce assembled communities with similar numbers of surviving species, which can be predicted using methods from the physics of disordered systems (refs. 28,29,36; Fig. 2b; Supplementary Note 3). In our analysis below, it will be convenient to treat the expected number of surviving species as an input parameter, and classify the assembled communities as a function of their niche saturation, $\mathcal{S}^*/\mathcal{R}$ (Supplementary Note 3.2). This re-parameterization allows for easier generalization to other community assembly models, since communities with similar levels of niche saturation will often have similar statistical properties, even if they resulted from different sampling distributions[29,35,36].

To account for evolution, we model the first mutational steps that occur in a randomly assembled community (Fig. 1b). This scenario might apply to the initial phases of in vitro passaging experiments[12,26], or recently colonized gut microbiomes[50,51]. Through much of our analysis, we will focus on a particularly simple class of "knock-out" mutations, where the mutant loses its ability to consume one of its resources ($r_i \to 0$). We assume that the cell can compensate for this deletion by increasing the uptake of other resources in its repertoire, but this compensation may not be perfect, corresponding to a shift in

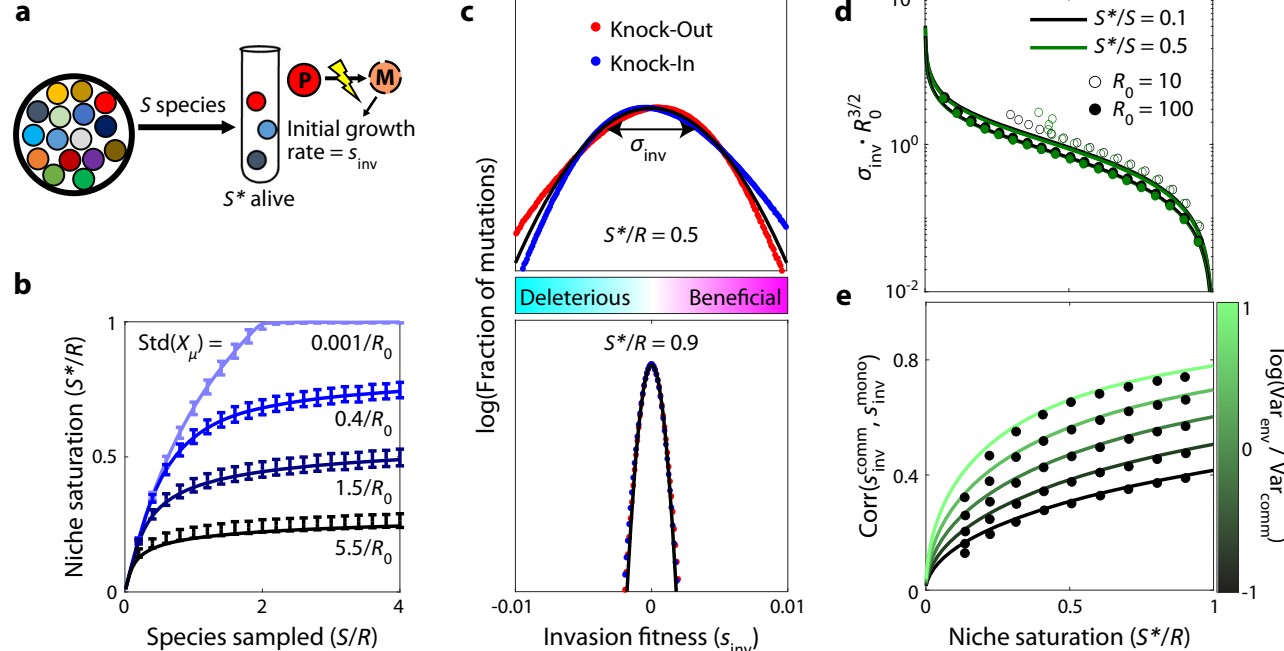

**Fig. 2 | Distribution of fitness effects of first-step mutations as a function of community complexity. a** As in Fig. 1, a community is assembled from $\mathcal{S}$ initial species, leaving $\mathcal{S}^*$ alive at equilibrium. The surviving species produce mutations, whose invasion fitness $s_{inv}$ is equal to their initial relative growth rate. **b** Number of surviving species $\mathcal{S}^*$ as a function of the sampling depth $\mathcal{S}$ and the standard deviation of the total uptake budget among sampled species, $Std(X_\mu)$. Curves show theory predictions from Supplementary Note 3.1, while points show means and standard deviations over $10^3$ simulation runs with $\mathcal{R} = 200$, $\mathcal{R}_0 = 40$, and uniform resource supply. **c** Distribution of fitness effects of knock-out (and knock-in) mutations with $\Delta X = 0$ in communities with two different levels of niche saturation ($\mathcal{S}^*/\mathcal{R}$). Black curve shows the theoretical predictions from Eq. (5), while dots

represent a histogram over all possible strategy mutations in $10^3$ simulation runs using the same parameters as panel (**b**), with $\mathcal{S}^*/\mathcal{S} = 0.1$. **d** The width $\sigma_{inv}$ of the distribution of fitness effects in panel C as a function of niche saturation, for various values of sampling permissivity $\mathcal{S}^*/\mathcal{S}$ and per-species resource usage $\mathcal{R}_0$. Curves show the theoretical predictions from Eq. (5), while the dots show the average over $10^3$ simulation runs. **e** Pearson correlation between the fitness effect of a mutation in the community and its fitness effect in monoculture, for different values of niche saturation and scaled variation in resource supply rates, $Var_{env}/Var_{comm} \equiv \left[Var(K)/\overline{K}^2\right] \cdot \mathcal{R}_0/(1 - \mathcal{R}_0/\mathcal{R})$. Curves show the theoretical predictions from Supplementary Note 4.1, while points show the average over all mutations in $10^3$ simulated communities with parameters the same as panel (**c**).

the effective uptake budget ($X \rightarrow X + \Delta X$). We also consider "knock-in" mutations, where a strain gains the ability to consume a resource (e.g., through horizontal gene transfer[25]), as well as more general changes that influence the uptake rates of multiple resources simultaneously ($\mathbf{r} \rightarrow \mathbf{r} + \Delta \mathbf{r}$). Following previous work[30,32], it will be convenient to classify these mutations by their impact on the strain's total resource uptake rate $X \equiv \log \sum_i r_i$ and its normalized resource consumption strategy $\boldsymbol{\alpha} \equiv \mathbf{r}/\sum_i r_i$, which describes the relative effort devoted to importing each of the resources. A general mutation will therefore involve a shift in one or both of these parameters ($X \rightarrow X + \Delta X$ and $\boldsymbol{\alpha} \rightarrow \boldsymbol{\alpha} + \Delta \boldsymbol{\alpha}$).

While the phenotypes of these mutants are simple, their fitness consequences will depend on the local environment, which is shaped by the other resident strains. Using the assembled communities as a backdrop allows us to quantify how this evolutionary landscape varies with the size and composition of the surrounding community.

### Distribution of fitness effects of mutations in newly assembled communities

The rate of evolution in a newly assembled community will depend on the supply of beneficial mutations. This landscape is often summarized by the local distribution of fitness effects (DFE), denoted by $\rho_\mu(s)$, which gives the relative probability that a mutation in focal strain $\mu$ will have an invasion fitness $s$. The shape and scale of $\rho_\mu(s)$ determine the availability of beneficial mutations, and therefore the rate of evolutionary change. Our resource competition framework allows us to ask how these features depend on the composition of the larger community.

For a community at ecological equilibrium, a mutation that arises in a resident strain $\mu$ and changes its resource uptake phenotype to a new value $X_\mu + \Delta X$ and $\boldsymbol{\alpha}_\mu + \Delta \boldsymbol{\alpha}$ will have an invasion fitness,

$$s_{inv} \approx \Delta X + \sum_i \Delta \alpha_i g_i, \tag{3}$$

where $g_i \equiv h_i/\overline{h} - 1$ is the excess availability of resource $i$ relative to the ecosystem average, $\overline{h} \equiv \frac{1}{\mathcal{R}}\sum_i h_i$[30] (Supplementary Note 1.5). Equation (3) shows that the invasion fitness of a mutation that only affects the overall uptake budget of a strain ($\Delta \boldsymbol{\alpha} = \mathbf{0}$) is independent of the surrounding community. In contrast, the benefits of mutations that change the resource consumption strategy of a strain will depend on the interactions between species, which are mediated by the values of the resource availabilities, $g_i$. For example, if a resource has a lower relative availability ($g_i < 0$), then it is not worth devoting a portion of its uptake budget to consume it, and a knock-out mutation for that resource should be beneficial.

Replica-theoretic calculations similar to ref. 36 allow us to predict the distribution of the $g_i$ for a typical assembled community when $\mathcal{R} \gtrsim \mathcal{R}_0 \gg 1$ (Supplementary Note 3.2). In large communities, the excess resource availabilities are well-approximated by a collection of Gaussian random variables,

$$g_i \sim \left(1 - \frac{\mathcal{S}^*}{\mathcal{R}}\right)\left[\frac{K_i}{\overline{K}} - 1 + Z_i \cdot C \cdot (\mathcal{S}^*/\mathcal{R})^{-1/2} \cdot \sqrt{\frac{1 - \mathcal{R}_0/\mathcal{R}}{\mathcal{R}_0}}\right], \tag{4}$$

where $\overline{K} \equiv \frac{1}{\mathcal{R}}\sum_i K_i$ is the average resource supply rate, $C$ is an $\mathcal{O}(1)$ factor that depends on the sampling depth $\mathcal{S}/\mathcal{S}^*$, and the $Z_i$ are uncorrelated standard Gaussians. Analogous expressions for other sampling distributions can be obtained by replacing the $\mathcal{R}_0$-dependent factor with the corresponding spread in $\boldsymbol{\alpha}$ (e.g., Supplementary Note 4.4). This mean-field solution requires that the resources are supplied at comparable rates ($|K_i - \overline{K}| \ll \overline{K}$) and each resource is utilized by a substantial number of surviving strains ($\mathcal{R}_0 \mathcal{S}^* \gg \mathcal{R}$; Supplementary Note 3.5; Supplementary Fig. 3d). When these conditions are satisfied, we can use Eqs. (3) and (4) to calculate the distribution of fitness effects $\rho_\mu(s)$ by aggregating over mutations with different values of $\Delta\boldsymbol{\alpha}$ and $\Delta X$.

To understand how the community influences the DFE of a focal species, it is helpful to begin by considering the simplest case, where the resource supply is uniform ($K_i = \overline{K}$) and mutations have no direct costs or benefits ($\Delta X = 0$). In this case, Eqs. (3) and (4) imply that $\rho_\mu(s)$ will also follow a Gaussian distribution with mean zero and standard deviation

$$\sigma_{\text{inv}} \sim \| \Delta\boldsymbol{\alpha} \| \cdot (1 - \mathcal{S}^*/\mathcal{R}) \cdot (\mathcal{S}^*/\mathcal{R})^{-1/2} \cdot \sqrt{\frac{1 - \mathcal{R}_0/\mathcal{R}}{\mathcal{R}_0}}, \quad (5)$$

where $\| \Delta\boldsymbol{\alpha} \| \equiv \sqrt{\sum_i \Delta\alpha_i^2}$ is the magnitude of the phenotypic change produced by the mutation. Examples for knock-out and knock-in mutations (where $\| \Delta\boldsymbol{\alpha} \| \approx 1/\mathcal{R}_0$) are shown in Fig. 2c–d. Since Eq. (5) does not explicitly depend on the parent strain $\mu$, it implies that the DFEs should be statistically similar for all strains in the community (Supplementary Fig. 2). Furthermore, since the details of the mutations only enter through their overall magnitude $\|\Delta\boldsymbol{\alpha}\|$, this implies that knock-out and knock-in mutations—as well as multi-resource mutations with the same value of $\|\Delta\boldsymbol{\alpha}\|$—will have statistically similar DFEs (Fig. 2c).

Equation (5) shows that the community influences the DFE of a focal strain primarily through the degree of niche saturation, $\mathcal{S}^*/\mathcal{R}$. The magnitude of the typical fitness effect approaches zero as $\mathcal{S}^*/\mathcal{R}$ increases (Fig. 2d), which is consistent with the idea that the rate of evolution will be slower in communities where more niches are already filled, since the establishment probability of a beneficial mutant is proportional to its fitness effect[52]. However, since the mean of the DFE is still centered at $s = 0$, the overall fraction of beneficial mutations remains constant as the niche saturation increases (Fig. 2c), even though a smaller number are likely to survive genetic drift. This implies that surviving organisms are not necessarily at a local evolutionary optimum, where any change to their resource consumption strategy tends to be deleterious.

However, this symmetry between the frequency of beneficial and deleterious mutations critically depends on the assumption of perfect trade-offs ($\Delta X = 0$), which will not always hold in practice. For example, a beneficial mutation could halt the production of an enzyme which is used for metabolizing a low-availability resource, while leaving the expression of other enzymes in that now-defunct pathway intact—resulting in a net cost to pure fitness. If all mutations carried such a direct cost ($\Delta X < 0$), then Eq. (3) implies that the entire DFE would shift to the left by a constant amount, $-|\Delta X|$. If this shift is larger than the typical spread of the DFE ($|\Delta X| \gg \sigma_{\text{inv}}$), then the beneficial tail of $\rho_\mu(s)$ will approach an exponential shape, whose height and width will both strongly decline with the degree of niche saturation $\mathcal{S}^*/\mathcal{R}$. Thus, the availability of beneficial mutations can sensitively depend on the genetic architecture of the strain's resource consumption rates.

Similar results apply when the resources are supplied at different rates ($K_i \neq \overline{K}$; Supplementary Note 4.1; Supplementary Fig. 7). In this case, the fitness effect of a mutation will depend on both the community context and the external environment, as encoded by the resource availabilities in Eq. (4). The overall magnitude of the environmental contribution declines as niche saturation $\mathcal{S}^*/\mathcal{R}$ increases,

consistent with previous work showing that larger communities self-organize to "shield" their member species from the external environment[30,32,36]. Interestingly, however, Eq. (4) shows that the *relative* impact of **K** on the fitness effect of a mutation actually increases with the degree of niche saturation, so the external environment can still exert an influence on the overall direction of the fitness landscape. This effect is strikingly illustrated when we compare the fitness effect of each mutation with its expected value in the absence of the community (Fig. 2e). We find that the correlations between the two DFEs can be substantial when the coefficient of variation in $K_i$ exceeds a critical value, $\sqrt{\mathcal{R}/\mathcal{S}^*\mathcal{R}_0} \lesssim 1$. This indicates that environmental shielding is often incomplete: the net direction of natural selection can be preserved across community backgrounds, even when interspecific competition exerts a strong effect on the local resource availabilities.

## Ecological diversification in large communities

While the invasion fitness of a mutation describes its initial growth rate, a successful variant will eventually reach a size where it starts to impact the other members of the community. Some of these mutants will eventually replace their parent strain, due to the principle of competitive exclusion[28]. Others can stably coexist with their parents by exploiting a different ecological niche[30]. How does the frequency of these in situ ecological diversification events depend on the composition of the surrounding community?

We can analyze the probability of mutant-parent coexistence in our model by recasting it as the outcome of two correlated assembly processes. First, an initial ecosystem $E_0$ is formed through our standard community assembly process (Fig. 1b, left). Then, a second ecosystem is formed when one of the surviving species in $E_0$ produces a beneficial mutant $M$, and the combined community $E_0 + M$ is allowed to reach its new ecological equilibrium (Fig. 1b, right). The latter event requires that the parent strain has a positive relative abundance in the first ecosystem ($f_P^{E_0} > 0$), and that the mutant survives in the second ecosystem ($f_M^{E_0 + M} > 0$). The probability that the mutant coexists with its parent can then be expressed as a conditional probability,

$$\mathbb{P}_{\text{coex}} = \mathbb{P}\left[ f_P^{E_0 + M} > 0 \,\Big|\, f_P^{E_0} > 0, f_M^{E_0 + M} > 0 \right], \quad (6)$$

which averages over the random resource uptake rates in the initial community, as well as the random effect of the adaptive mutation.

The correlated assembly process in Eq. (6) is challenging to analyze with existing methods[37], since the mutant and parent phenotypes are closely related to each other. Fortunately, we will show that we can often approximate the coexistence probability by considering a third community assembly process, in which the mutant and parent are introduced simultaneously with the other species. This "simultaneous assembly" approximation differs from the two-ecosystem model in Eq. (6), since the final community can contain "rescued" species that would not have survived in $E_0$ before the mutant strain invaded. However, simulations indicate that these differences result in only small corrections to the coexistence probability across a broad range of parameters (Supplementary Note 2.1, Supplementary Fig. 4), so this is often a reasonable approximation.

When the simultaneous assembly approximation holds, we can evaluate the coexistence probability by extending the replica-theoretic calculations used for Eq. (4) (Supplementary Note 3.3). We find that coexistence can be expressed as the probability that the invasion fitness of the mutant lies below a critical fitness threshold,

$$s_{\text{coex}}(f_P) \equiv \sigma_{\text{inv}} \cdot (f_P \mathcal{S}^*) \cdot (\mathcal{S}^*/\mathcal{R})^{-1/2} \cdot \frac{\sqrt{2} \, \| \Delta\boldsymbol{\alpha} \|}{\| \Delta\boldsymbol{\alpha}_{\text{comm}} \|}. \quad (7)$$

where $f_P$ is the relative abundance of the parent strain, $\| \Delta\boldsymbol{\alpha} \|^2 \equiv \sum_i \Delta\alpha_i^2$ is the phenotypic effect size of the mutation, and

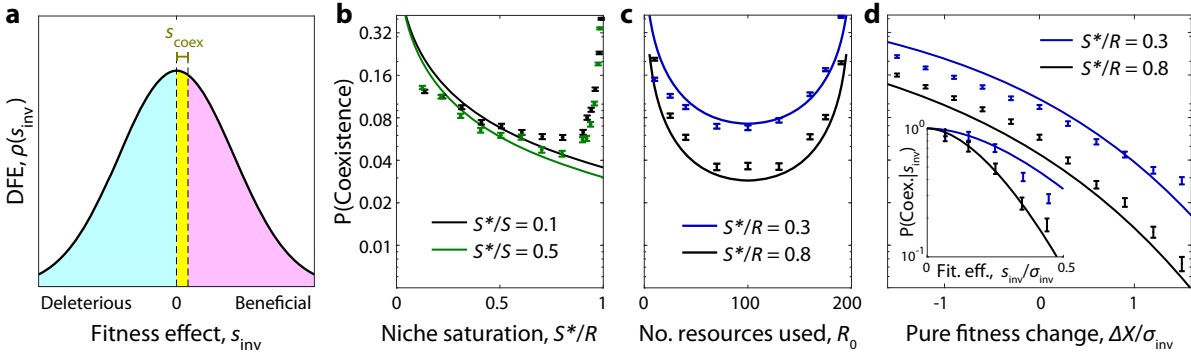

**Fig. 3 | Ecological diversification in large communities. a** Schematic showing the subset of mutations that are able to stably coexist with their parent strain. Coexistence occurs in the yellow region where the invasion fitness is below a critical threshold $s_{coex}$, which depends on the abundance of the parent and the phenotypic effect size of the mutation. **b–d** Probability that a successful knock-out mutant coexists with its parent strain as a function of (**b**) the niche saturation $\mathcal{S}^*/\mathcal{R}$, (**c**) the typical number of resources used per species $\mathcal{R}_0$, and (**d**) the change in overall uptake budget of the mutant $\Delta X$. Inset shows the dependence on the total invasion fitness $s_{inv}$. In all panels, curves show the theory predictions from Supplementary Notes 3.3 and 3.4, while points show means and standard errors over $10^4$ simulation runs with base parameters $\mathcal{R} = 200$, $\mathcal{R}_0 = 40$, $\mathcal{S}^*/\mathcal{S} = 0.1$, and uniform resource supply.

$\| \Delta\boldsymbol{\alpha}_{comm} \|^2 \approx 2(1 - \mathcal{R}_0/\mathcal{R})/\mathcal{R}_0$ is the equivalent spread between random pairs of strains in the community. Averaging over the abundance of the parent strain (Supplementary Note 3.4) yields a corresponding expression for the coexistence probability as an integral over the DFE in Fig. 2c,

$$\mathbb{P}_{coex} \approx \frac{\int_0^\infty s\rho(s)e^{-s/\bar{s}_{coex}}}{\int_0^\infty s\rho(s)ds} \qquad (8)$$

where $\bar{s}_{coex}$ is the coexistence threshold for a typical genetic background ($f_P \sim 1/\mathcal{S}^*$). For mutations with modest phenotypic effects, such as a single resource knock-out ($\| \Delta\boldsymbol{\alpha} \| \sim 1/\mathcal{R}_0$), the magnitude of $\bar{s}_{coex}$ will be much smaller than the typical spread in the DFE in Fig. 2c. This implies that only mutations with anomalously low invasion fitnesses will have an appreciable chance of coexisting with their parent (Fig. 3a).

For a mutation with perfect trade-offs ($\Delta X = 0$), the integral in Eq. (8) reduces to a simple form,

$$\mathbb{P}_{coex} \approx \frac{1}{\mathcal{S}^*/\mathcal{R}} \cdot \frac{2 \| \Delta\boldsymbol{\alpha} \|^2}{\| \Delta\boldsymbol{\alpha}_{comm} \|^2}, \qquad (9)$$

which depends on the niche saturation $\mathcal{S}^*/\mathcal{R}$ and the phenotypic effect size of the new mutation.

Equation (9) shows that the coexistence probability is largest for small values of $\mathcal{S}^*/\mathcal{R}$ (Fig. 3a), consistent with the idea that diversification is easier when there are many open niches left to exploit, as in an adaptive radiation[20]. Interestingly, however, we find that the coexistence probability does not vanish as the community approaches saturation ($\mathcal{S}^*/\mathcal{R} \to 1$), but instead plateaus at a nonzero value. This is true even in fully saturated communities ($\mathcal{S}^* = \mathcal{R}$), where other species must be driven to extinction when the successful mutant invades.

In this regime, the coexistence probability is most strongly determined by the phenotypic effect size of the mutation. A mutation that changes the resource consumption strategy by an infinitesimal amount ($\| \Delta\boldsymbol{\alpha} \| \to 0$) can never coexist with its parent, while a mutation with $\| \Delta\boldsymbol{\alpha} \| \sim \| \Delta\boldsymbol{\alpha}_{comm} \|$ has a much higher probability of coexistence. Single-resource knockouts fall between these two extremes, with the coexistence probability scaling inversely with $\mathcal{R}_0$ (the average number of resources utilized per strain), rather than the total number of resources consumed by the community. This suggests that in situ ecological diversification can continue to occur even in large

communities containing many species and resources (Fig. 3b and Supplementary Fig. 5).

We see that simulation results generally support the theoretical predictions in Eq. (9), but start to exceed this value for communities very close to full saturation. The source of this deviation coincides with the breakdown of our "simultaneous assembly" approximation above, which allowed the invading mutant to "rescue" some species that had previously gone extinct. The deviations disappear if extinct species are allowed to re-invade (Supplementary Fig. 4), suggesting that competition from rescued species plays a significant role in our theory for extremely high levels of niche saturation. This rescuing effect could be relevant in some natural settings, e.g., if the species pool is maintained in a separate spatial niche with frequent migration back into the community. Regardless, our results in Eq. (9) provide a lower bound on the coexistence probability in large ecosystems, suggesting that diversification could be even more common under some conditions.

In reality, most mutations that change an organism's resource consumption strategy are unlikely to be perfect trade-offs. More generally, we find that mutations that carry a direct cost or benefit $\Delta X$ change the coexistence probability in Eq. (9) by the factor

$$\frac{\mathbb{P}_{coex}(\Delta X)}{\mathbb{P}_{coex}(\Delta X = 0)} \sim \begin{cases} \frac{\sigma_{inv}}{\sqrt{2\pi}\Delta X} e^{-\Delta X^2/2\sigma_{inv}^2} & \text{if } \Delta X \gg \sigma_{inv}, \\ 1 & \text{if } |\Delta X| \ll \sigma_{inv}, \\ \Delta X^2/\sigma_{inv}^2 & \text{if } \Delta X < 0, |\Delta X| \gg \sigma_{inv}. \end{cases} \qquad (10)$$

which shows that direct costs or benefits begin to exert an effect when $\Delta X$ becomes comparable to $\sigma_{inv}$. However, the direction of this effect is somewhat counterintuitive: strategy mutations that carry a direct fitness cost ($\Delta X < 0$) are more likely to coexist with their parent, provided that their overall invasion fitness is still positive. Conversely, mutations with strong direct benefits ($\Delta X > 0$) are more likely to drive their parent strain to extinction (Fig. 3b). These differences arise because nonzero values of $\Delta X$ change the typical invasion fitnesses of successful mutants. The coexistence probability in Eq. (8) is highly sensitive to these shifts, as mutants with an invasion fitness below $\bar{s}_{coex}$ are much more likely to coexist with their parent (Fig. 3a, d). This is a reflection of the fact that diversification is driven by a narrow range of mutants with fitness high enough to establish but small enough to coexist.

We can gain some intuition for this effect by considering a loss-of-function mutation for a resource that is currently overutilized by the population ($g_i < 0$). Equation (2) shows that the growth rate of this variant (relative to its ancestor) is given by $\sim g_i(t)$. As the mutant grows

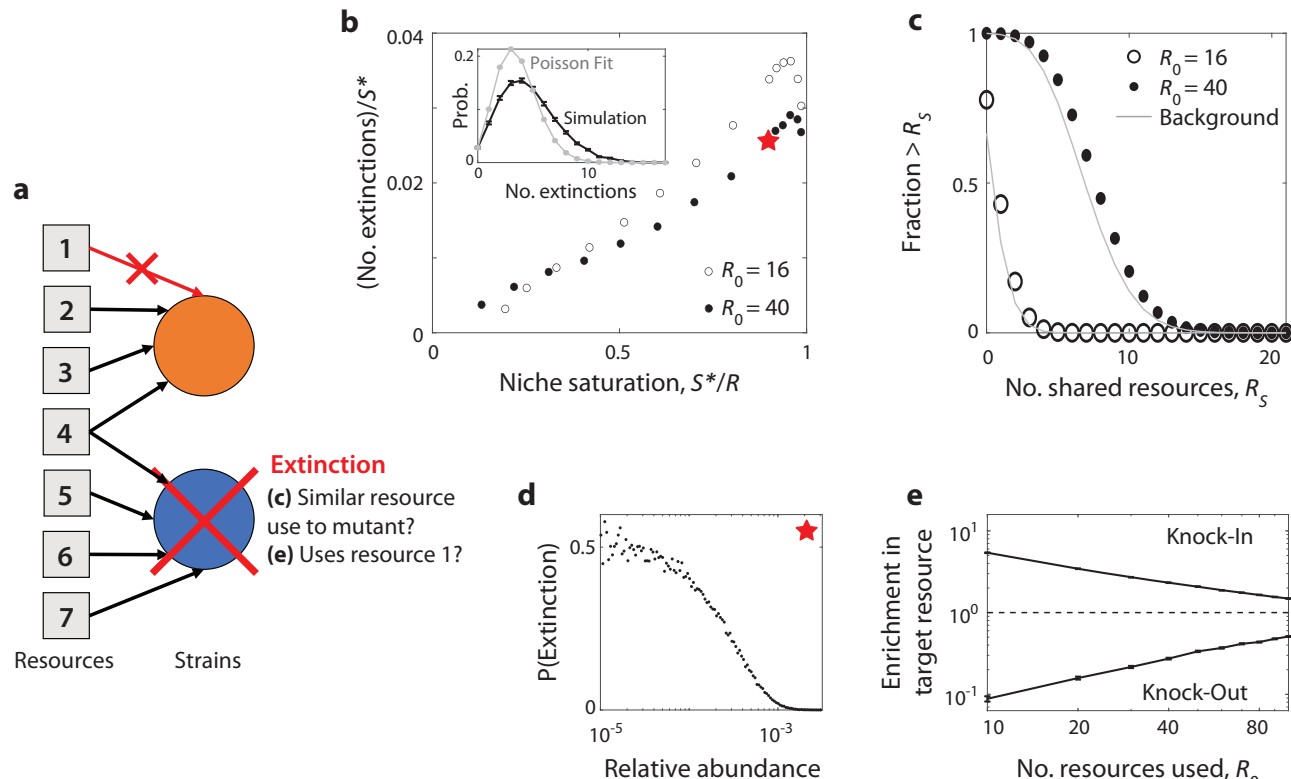

**Fig. 4 | Successful mutations drive extinctions of metabolically distant species.**
**a** Schematic showing the extinction of an unrelated species (blue) after a beneficial knockout mutation (orange) invades. In this example, the displaced species and the mutant share one common resource, but not the one targeted by the knock-out mutation. **b** Average number of extinctions after the invasion of a successful mutant, as a function of niche saturation $S^*/\mathcal{R}$; parent strains are excluded from the extinction tally. Inset: full distribution of extinctions for the starred parameters, compared to a Poisson distribution with the same fraction of zero counts. Points denote the averages over $10^4$ simulation runs with the same base parameters as Fig. 3. **c** Distribution of the number of resources jointly utilized by the displaced species and the invading mutant (parent strains excluded). Points denote the results of simulations with $S^*/\mathcal{R} = 0.9$. Gray curves show the analogous background distribution between the mutant and all other species in the community, regardless of whether they become extinct. **d** Probability of extinction as a function of initial relative abundance for the starred point in panel (**b**). **e** The fold change in probability that the displaced species uses the same resource targeted by the mutant (i.e., the resource being knocked in or out), relative to the background distribution of resource use in the population. Points denote means and standard errors for knock-in and knock-out mutations as a function of $\mathcal{R}_0$ for $S^*/\mathcal{R} = 0.8$, with the the remaining parameters the same as panel (**b**).

in abundance, it begins to replace its parent, which tends to reduce the overall utilization of the target resource ($\partial_t g_i > 0$). If $g_i$ reaches zero before the parent strain goes extinct, then the mutant will coexist with its parent, having lost its initial growth advantage. This coexistence is most likely to happen if the parent strain has high abundance, or if $|g_i|$ (and therefore $s_{inv}$) is initially small (Supplementary Fig. 6).

The same argument applies for more complex mutations which affect multiple resources (Supplementary Fig. 5). It also explains why mutations with small phenotypic effects (e.g., a pure fitness mutation with no strategy change) cannot coexist with their parent. If the mutant and parent have near-identical resource consumption strategies, the mutant's invasion produces very little negative feedback on its growth rate relative to its parent, making it likely the parent strain will be driven to extinction (Supplementary Fig. 5b). Similar logic applies for extensions of our basic model, including non-uniform resource supply rates and alternative community assembly schemes (Supplementary Figs. 3, 8 and 9; Supplementary Note 4). Together, these results suggest that in situ diversification could be common even in large and saturated communities, particularly for mutants on abundant backgrounds with lower-than-expected invasion fitnesses.

**Successful mutations drive extinctions in other niches**
In addition to displacing their parents, successful mutants can also drive other species in the community to extinction. This is particularly

evident in saturated communities ($S^* = \mathcal{R}$), where competitive exclusion implies that the invasion of a new strain must be accompanied by extinction of at least one other. Figure 4b shows that the number of extinctions steadily rises with the degree of niche saturation, exceeding 1% of the initial community ( ~1–10 species) for many combinations of parameters. Moreover, these extinctions are not completely independent of each other, since they are somewhat overdispersed compared to a simple Poisson expectation (Fig. 4b, inset).

Past analyzes of microbial communities have suggested that phylogenetically distant strains could exhibit correlated dynamics if they have similar resource consumption strategies[26]. If this hidden metabolic similarity drives the extinctions in our model, we would expect the species that go extinct after mutant invasion to be more metabolically similar to the mutant than a typical species in the community. We tested this idea by examining the number of resources that were jointly consumed by the invading and displaced strains (a proxy for their overall metabolic similarity). Interestingly, we found that the number of resources shared with the mutant was not substantially higher for the displaced species, and was comparable to a randomly drawn species from the larger community (Fig. 4c). This suggests that the extinction events in Fig. 4 cannot be explained by traditional measures of niche overlap[33]. Rather, successful mutants can displace species outside of their apparent niche, even when they stably coexist with a more metabolically similar parent.

Since extinctions were not well-predicted by their overall metabolic similarity to the mutant, we conjectured that these displaced species may possess other features that render them vulnerable to extinction. For example, low-abundance species may be less well-adapted to the current environment, and thus sensitive to perturbations like the invasion of a mutant. Consistent with this hypothesis, we found that the extinction probability for very low-abundance species is much higher than the community average, approaching ~50% at the highest levels of niche saturation (Fig. 4d). Furthermore, although the displaced species are metabolically distant from the invading mutant, we find they are more likely to share the mutant's strategy for the resource targeted by the mutation. Displaced species are more likely to use the resource gained by a successful "knock-in" mutation, and are correspondingly less likely to use the resource lost in a successful knock-out strain (Fig. 4e). These results illustrate that in high-dimensional ecosystems, the invasion of a new mutant can have a small impact on a diverse range of metabolic strategies. If a resident strain is maladapted enough to already be on the edge of extinction, the invasion of a mutant can be enough of a perturbation to displace it from the community. Thus, the abundance of an organism might often be a better predictor of its fate than its apparent metabolic niche.

## Robustness of ecosystem to subsequent mutations

Our analysis above focused on the first wave of mutations arising in a newly assembled community, where the initial states could be predicted using existing community assembly theory[28,29,35,36]. However, subsequent waves of evolution could eventually drive the ecosystem away from this well-characterized initial state[30]. To assess the robustness of our results under the acquisition of further mutations, we simulated successive waves of mutations using a generalization of the approach in Fig. 3. We considered the simplest case where resident strains could generate knock-out mutations on any of the resources they currently utilized. We also assumed that the dynamics were mutation-limited[30,31], so that the community relaxes to a well-defined steady state between each successive mutation. We continued this process until one of the surviving strains had accumulated 10 mutations in total, which typically corresponded to 100–200 successful mutations in the larger community.

We first asked how these subsequent waves of mutations altered the genetic structure of their community. While the total number of surviving strains decreased slightly over time (eventually stabilizing at an intermediate value), the number of strains related through in situ diversification events increased approximately linearly over the same time window (Fig. 5b). This indicates that the ecological diversification events in Fig. 3 continue to occur at a high rate even after additional mutations have accumulated, more quickly than individual branches of diversified lineages go extinct. Nonetheless, the abundance trajectories in Fig. 5a indicate that these extinction events among close relatives are not uncommon. For example, Lineage 1 seeds eight ecological diversification events over the course of the simulation, but only three of these strains are alive at the end of the simulation. On average, we find that newly diversified lineages coexist for ~40 mutational steps before one of them goes extinct (Fig. 5c). Nonetheless, longer coexistence is possible: one of the diversification events in Fig. 5a is maintained for >150 mutational steps, enough time for multiple additional mutations to accumulate within each lineage. Thus, coexisting mutant-parent pairs are often maintained through further evolutionary perturbations, even as the total number of species stabilizes over time.

Our first-step analysis showed that mutations in more abundant strains were more likely to coexist with their parent. This prediction is borne out by our multi-step simulations as well: while some highly abundant lineages seeded many in situ ecological diversification events (e.g., Lineage 1 in Fig. 5a), the typical surviving lineage experienced no more than one (e.g., Lineage 2), and many species went extinct without diversifying at all (e.g., Lineage 3). Consistent with our

earlier predictions, lineages at low abundance were more prone to extinction when a new mutant invaded, while coexistence events tended to happen to high-abundance lineages (Fig. 5a, right). The combination of these factors creates a "rich get richer" effect where high-abundance species diversify at the expense of driving low-abundance species to extinction[30].

Finally, we asked how the accumulation of mutations shifts the landscape of beneficial mutations from the initial "assembled" state in Figs. 2 and 3. We found that the distribution of fitness effects does not dramatically differ between the start and the end of the simulation, although its shape changes slightly from a Gaussian distribution toward a two-tailed exponential (Fig. 5d). In contrast, the impacts of successful mutations exhibit larger changes: the proportion of mutation events which result in mutant-parent coexistence increases substantially with evolutionary time, rising from about 20% to 80% over the course of the simulation (Fig. 5e). This shift cannot be explained by changes in the overall number of species or the availability of beneficial mutations, but instead deviates entirely from the replica-theoretic predictions that describe our initial state. Thus, while many of our qualitative conclusions continue to hold on longer evolutionary timescales, the accumulation of mutations can lead to quantitative differences from our first-step analysis above. In both cases, the evolved communities exhibit distinct genetic signatures compared to their purely assembled counterparts, which could motivate future tests of in situ evolution.

## Discussion

Large ecological communities apply complex evolutionary pressures to their resident species, leading to controversy about how these organisms' evolutionary trajectories are affected by the background community[17]. Here, we address this challenge by developing a theoretical framework for predicting the first steps of evolution in large, randomly assembled communities that compete for substitutable resources. These "mean-field" results hold when surviving strains consume many resources, and each resource is consumed by many surviving strains. Under these conditions, our results provide a mechanistic approach for understanding how the fitness benefits and fates of new mutations should scale with the diversity and metabolic overlap of the surrounding community.

Our results show that the supply of beneficial mutations does not necessarily run out in larger communities – as expected in the simplest models of niche filling – but rather that the benefits of these mutations will systematically decline with the degree of niche saturation ($S^*/\mathcal{R}$). We also find that the fitness effects of mutations can be broadly correlated with the external environment, even in large communities where internal resource concentrations are broadly shielded from external environmental shifts[32,36]. These distributions of fitness effects can be measured in modern experiments, e.g., by performing bar-coded fitness assays in communities of different size[12,50].

Our finding that successful mutants often coexist with their parents is reminiscent of empirical observations from the gut microbiome, where recently diverged strains differing by only a handful of mutations appear to stably coexist within their host[5,6]. While spatial structure could also contribute to this coexistence[27,53], our model demonstrates that in situ diversification can continue to occur even in a well-mixed environment when most niches are already filled. Similar behavior has also been observed in generalized Lotka-Volterra models with spatiotemporal chaos[54]. This suggests that ongoing diversification may be a generic feature of large microbial communities, providing an alternative mechanism for the "diversity begets diversity" effect[17,55,56] that does not rely on explicit cross-feeding interactions.

Beyond diversification, Fig. 5 demonstrates that successful mutants can also drive distantly related species to extinction. This behavior is consistent with recent empirical observations in the human gut microbiome[8] and strain-swapping experiments in synthetic gut

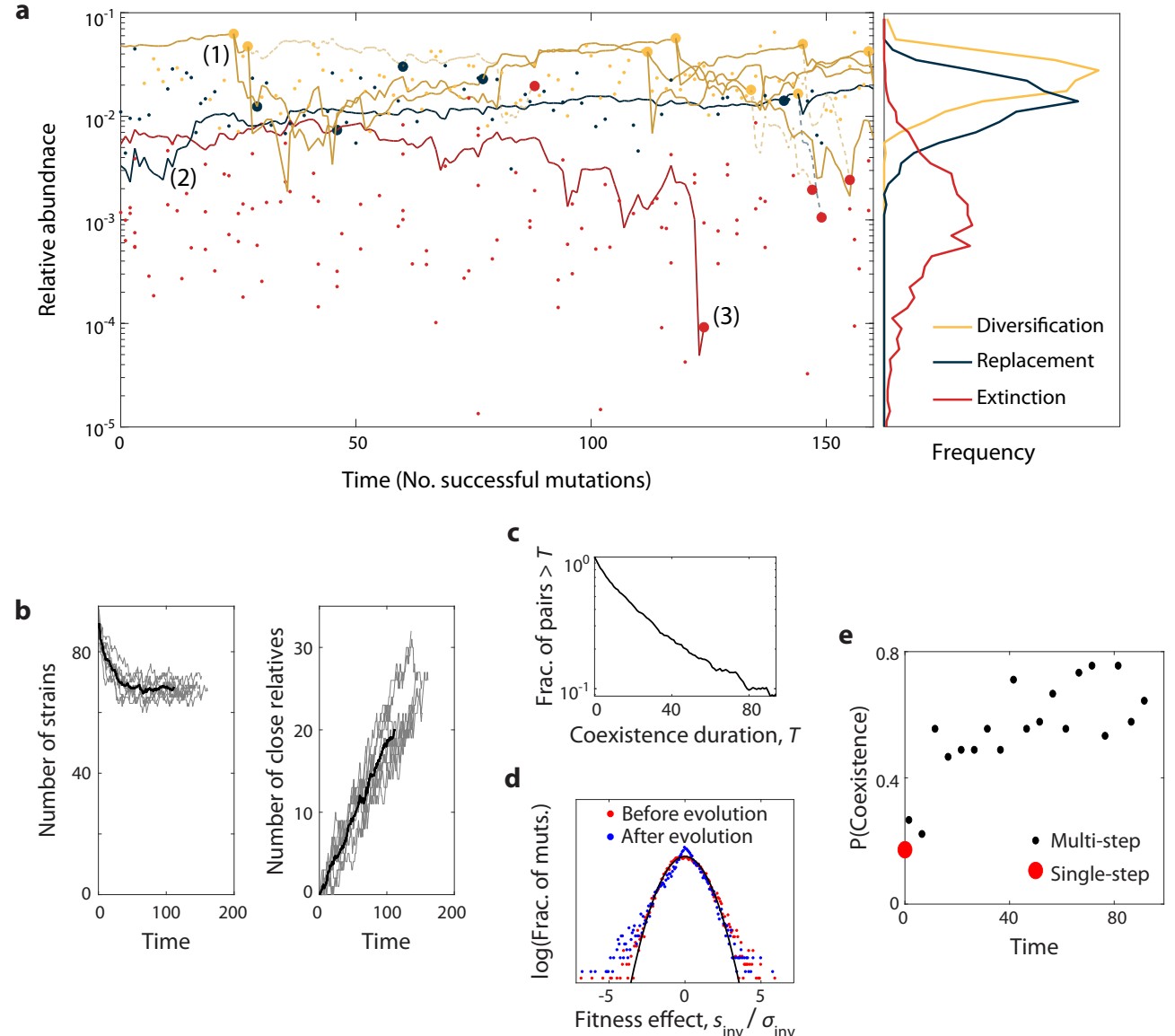

**Fig. 5 | Mutation and diversification over longer evolutionary timescales. a** Left: an example simulation showing the step-wise accumulation of ~100 adaptive knock-out mutations in a community with $\mathcal{R} = 100$, $\mathcal{R}_0 = 20$, $\mathcal{S}^*/\mathcal{R} = 0.9$, $\mathcal{S}^*/\mathcal{S} = 0.1$. Solid lines denote the abundance trajectories of 3 example lineages. Large points indicate extinction events in these lineages (red), diversification events (yellow), and mutation events that displace their parent strain (blue) in the highlighted lineages, while smaller points indicate analogous events for other species in the community. Dashed lines illustrate offshoots of the highlighted lineages that eventually went extinct. Right: Relative abundances of strains when they experienced mutation, diversification and extinction events, respectively. Lines denote histograms aggregated over 9 simulation runs. **b** Left: Total number of surviving strains over time. Gray lines denote replicate simulations for the same parameters in panel (**a**), while their average is shown in black. Right: Total number of strains related to another surviving strain through one or more in situ diversification events. **c** The probability of mutant-parent coexistence being maintained (i.e., both lineages surviving) as a function of time since initial divergence, over ten simulations. **d** The distribution of fitness effects at the start of the simulation (red) and after 90 accumulated mutations (blue), compared to the first-step predictions from Fig. 2 (black). **e** The probability that a mutation event leads to stable diversification as a function of time. Black points denote binned values aggregated over nine replicate simulations, while red point denotes the analogous result for $10^4$ first-step simulations.

communities[57]. It could also contribute to the extinction events observed in community passaging experiments[58]. The fact that the displaced species are metabolically diverged from the invading mutants creates obstacles for inferring these interactions from metabolomics measurements[59] or metabolic reconstructions[60]. Our findings suggest that future efforts should instead focus on how the invading mutant impacts low-abundance species more generally, and whether they utilize the specific resource(s) targeted by the mutation. These results align with recent work emphasizing the importance of collective interactions in shaping microbial community dynamics[61].

Here, we focused on the simplest possible resource competition model, neglecting important factors like spatial structure[62], metabolic regulation[46], and more general forms of cross-feeding[49,63], which are all thought to play key roles in natural microbial communities. We also considered the simplest regime of evolutionary dynamics that neglect competition between simultaneously occurring mutations. These clonal interference effects can enhance diversity by allowing strains to temporarily evade competitive exclusion[31]. On the other hand, competition between lineages also selects for more strongly beneficial mutations[64],

which we predict are less likely to coexist with their parent. Our results provide a baseline for incorporating these effects, which will be crucial for understanding how large microbial communities will evolve.

## Methods

See Supplementary Note 1 and the first part of the Results section for a description of the theoretical framework used throughout, and Supplementary Notes 2–4 for derivations of our results from this model. We detail the simulations used to validate our conclusions in Supplementary Note 5.

## Data availability

Simulation results and processing code used to generate figures can be found on Zenodo[65].

## Code availability

Source code for community assembly simulations, numerical calculations and figure generation are available on Zenodo[65] as well as Github (github.com/jdmcenany/First_Step_Muts).

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

## Acknowledgements

We thank Daniel Fisher for useful discussions, and Sophie Walton, James Ferrare, Zhiru Liu, Daniel Wong, and Avaneesh Narla for comments and feedback on the manuscript. This work was supported in part by the Alfred P. Sloan Foundation (FG-2021-15708), NIH NIGMS Grant No. R35GM146949, and a Terman Fellowship from Stanford University (B.H.G.). B.H.G. is a Chan Zuckerberg Biohub - San Francisco Investigator.

## Author contributions

Conceptualization: J.M. and B.H.G.; Theory and methods development: J.M. and B.H.G.; Analysis: J.M. and B.H.G.; Writing: J.M. and B.H.G.

## Competing interests

The authors declare no competing interests.
