## [Peer Review File · Nature Communications]

Predicting the First Steps of Evolution in Randomly Assembled CommunitiesREVIEWER COMMENTS

Reviewer #1 (Remarks to the Author):

The manuscript "Predicting the first steps of evolution in randomly assembled communities" presents an extensive statistical analysis of the fate of mutants and their effect on resident populations that survived competing for numerous substitutable resources.

It includes both the analytic predictions for various averages, probabilities and correlations and the simulation results that mostly confirm the analytics. The work appears to be an extension of Ref. (30) to a large number of resources and coexisting species, which required a different theoretical approach.

Specifically, analytical predictions based on technique, first applied to ecological equilibria in (36,37), are made for the statistics of the number of coexisting species, invasion fitness of first mutants, correlation between invasion fitness for various levels of diversity of residents, the effect of mutation fixation on its ancestral and other species survival, etc. Most of analytic predictions fit common sense and were confirmed by numerically-produced statistics. While the details of derivation may appear hard to follow for a general specialist in ecology and evolution, the concordance between theory and numerics gives a "proof of the principle" for applicability of the replica methods to problems of random community assembly and invasibility. At the same time, the utility of analytic estimates appears to be limited to specific probability distributions of uptake rates and resource inputs, which is in contrast to conceptually simple and broadly useful numerical simulations.

Overall, the study, albeit limited mostly to the scenarios with early mutations, is very thorough and detailed. At the same time, the presentation style leaves room for improvement. In the present form the manuscript does not appear reader-friendly to the general audience of Nature Communications.

I would like to suggest:

1. To specify resource dynamics in the main text
2. Eq. 1 needs to be first written in original, intuitively clear, terms such as $r_{\{\mu, i\}}$. Only after, if necessary for further analysis, it could be encoded in α_j 's and exponentiated logarithm. Apparently, the opening statement closely resembles that in Ref. (30), but that itself does not prove its

suitability.

3. Do upper case K_i in Figure 1 denote the same quantity as lower case k_i in the definition of h_i below Eq. 1.

4. What is R_0 ? It's hard to get what the symbol " \sim " means here.

5. It is necessary to explain why γ is defined proportional to R_0 ? Once again, it would be easier on a reader to retain the original notations ($r_{\mu,i}$) where possible in the main text and use the derived ones for replica integration, etc.

6. "For example, if a resource has a lower relative availability ($g_i < 0$), then it is not worth devoting energy to consume it, and a knock-out mutation for the resource should be beneficial." I don't think this energy is accounted for in the model.

7. What is $\delta \vec{k}$?

8. The term ϵ in Fig. 2 needs to be defined

9. "However, since the mean of the DFE is still centered at $s = 0$, the overall fraction of beneficial mutations remains constant as $S^* \rightarrow R$ (Fig. 2C)." I find this statement rather misleading since the probability of fixation, usually estimated as the fraction of invasion fitness and the birth rate, becomes negligible when $S^* \rightarrow R$.

10. Fig. S5 "Dashed line" -> blue line?

11. Once again, I suggest to move a large part of section 1.1 of Supplementary materials to the main text.

12. To prove that all approximations are applicable, it would be illustrative to show that, for example, Fig. 2B is reproducible via direct integration of equations for population and resource dynamics. A self-consistency check that the consumption terms are always much larger than the dilution rate would also be convincing.

13. Eqs. 6-8 are unreadable in the main text. For example, I failed to figure out how for " $R_0 \gg 1$ ", the integral in Eq. (6) further reduces to eq. 8". I suggest to leave them in Supplementary materials and present the resulting Eq. (8), to illustrate the verbal explanations of various regimes.

14. I could not see how (S4) is derived from (S1-S3)

15. Lines 707-708 This has been known long before (30,32), see, for example, (45)

Reviewer #2 (Remarks to the Author):

McEnany and Good study a resource-competition model with phenotypic mutations and describe some properties of random ecosystems invaded by such mutants. A key question that the authors ask is whether new mutants generally knock-out or coexist with their parent. They show that mutants can generally coexist with their parents, instead knocking out other metabolically distant species. The species most likely to go extinct are the ones at low abundance. Though the authors' calculations pertain only to the first steps of this eco-evolutionary process of mutants invading and modifying the community, the authors show numerical results suggesting that these general lessons tend to hold even after many steps of eco-evolution.

The paper is well-written and addresses a very interesting topic in the field. Eco-evolutionary dynamics have largely remained difficult to study from the ecological direction. Most population genetics assume that the rest of the community is implicit.

However, our chief concern is that the authors derive their results in a very specific regime of the model where all species are effectively neutral (detailed below), which raises questions about the generality of the results. We believe the authors should address this assumption in the main text, and appropriately tone down their claims. Below, we detail our chief concern as well as highlight a few others, which we recommend the authors to address in a revised version of the manuscript.

MAJOR COMMENTS

- The main assumption that allows the authors to calculate the key quantities of interest (e.g., \mathbb{P}_{coex}) is stated in section 1.2 of the SI. The authors decompose $\tilde{h}_i = 1 + g_i$ and Taylor expand $\tilde{F}(g_i)$ for small g_i . This decomposition (and retaining only $g_i + g_i^2$ terms) implicitly assumes that rescaled resource availability for any resource is 1 near equilibrium, forcing the system to stay near a nearly neutral point where all resource availabilities are ≈ 1 . This effectively makes all species fitnesses $X_\mu \approx 1$, making them nearly neutral. This assumption is what allows the authors to frame their problem as an optimization problem, to which replica-theoretic techniques can be applied.

This is a rather strong assumption that makes the analysis non-generic in our opinion, and that to our knowledge is not discussed in the main text. We suggest the authors explicitly mention this in the abstract, results, and/or discussion, since it is not clear if their results hold outside of this nearly neutral regime.

- Though the presentation is clear in most of the places, there are few places where it could be clearer. For example:

--- The physical meaning of D (in S57) could be explained in more words. In section 3.3 (scaling analysis of \mathbb{P}_{coex}), the authors state that $D > E$ represents the requirement that the

mutant is well-adapted enough to survive in the population. We could see it somewhat, but it could be made more apparent.

--- We can somewhat see that S87 represents the relative abundance of parent species (as $\frac{2}{SI'(\lambda)} \approx \frac{1}{S^*}$ in large $|\lambda|$ limit) but not exactly.

MINOR COMMENTS

- Since $\sum_{\mu=1}^P n_{\mu}(t) = \sum_{i=1}^R \kappa_i + \text{Big}(\sum_{\mu=1}^P n_{\mu}(t) - \sum_{i=1}^R \kappa_i) e^{-t}$. Dividing by $\sum_{i=1}^R \kappa_i$ and $\sum_{\mu=1}^P n_{\mu}$ are not the same thing in equation S2.

- Equations S6a and S6b don't seem quite right if in S6b, the second term in the brackets is meant to represent the unutilized resources. Because if l_i is the fraction of i^{th} resource that is utilized by some species, then it is $(1-l_i)$ that is excreted back to the environment.

- In section 2.5 (Parent-Mutant Coexistence Probability), authors have only considered the global minimum for $e^{-\frac{1}{2} \vec{v} \mathbf{M}^{-1} \vec{v}^t}$, whereas there exist cases in which global minimum of this function, can lie outside the non-positive-resource-surplus region and still have a point in the constrained region for which the function is optimum.

Detailed Response to Reviewer Comments

Note: line numbers refer to the track-changes version of the manuscript. We have highlighted the substantive changes in red, while minor grammatical edits were omitted for readability.

Reviewer #1 Comments:

The manuscript "Predicting the first steps of evolution in randomly assembled communities" presents an extensive statistical analysis of the fate of mutants and their effect on resident populations that survived competing for numerous substitutable resources. It includes both the analytic predictions for various averages, probabilities and correlations and the simulation results that mostly confirm the analytics. The work appears to be an extension of Ref. (30) to a large number of resources and coexisting species, which required a different theoretical approach. Specifically, analytical predictions based on technique, first applied to ecological equilibria in (36,37), are made for the statistics of the number of coexisting species, invasion fitness of first mutants, correlation between invasion fitness for various levels of diversity of residents, the effect of mutation fixation on its ancestral and other species survival, etc. Most of analytic predictions fit common sense and were confirmed by numerically-produced statistics. While the details of derivation may appear hard to follow for a general specialist in ecology and evolution, the concordance between theory and numerics gives a "proof of the principle" for applicability of the replica methods to problems of random community assembly and invasibility.

We thank the reviewer for their positive comments and overview of our work.

(1) At the same time, the utility of analytic estimates appears to be limited to specific probability distributions of uptake rates and resource inputs, which is in contrast to conceptually simple and broadly useful numerical simulations.

Thanks for bringing this up – while it might not be obvious at first glance, our analytical results actually hold for a much broader range of uptake rate distributions and resource inputs than the specific example we focused on in the Main Text. The reasons for this are closely related to the central limit theorem – it applies in our case because in a large community, each resource will typically be consumed by multiple surviving strains, while each strain will subsist on multiple resources.

However, we agree that we did not make this point sufficiently clear in the previous version of the manuscript, and we appreciate the Reviewer's suggestion that numerical simulations can provide an accessible way to demonstrate this point for a broader audience. We have therefore revised our manuscript to incorporate numerical simulations for a variety of additional scenarios – including some designed to test the limits of validity of our analytical derivations. In particular, we have added additional figures that consider:

(i) Higher variation in resource supply (exponentially distributed supply rates, Fig. S7B-C). We find that even though our theoretical analysis assumes that the mean resource supply is much greater than the spread, simply plugging in this larger variance for communities with a given niche saturation S^*/R predicts the DFE and coexistence probability reasonably well.

(ii) Wider variation in the number of resources utilized by each strain (Fig. S9). We find that this sampling process can lead to communities with both more specialized and more generalist

resource profiles, but does not have unexpected effects on the short-term evolutionary dynamics we study.

(iii) Different forms of metabolic trade-offs between the number of utilized resources and the maximum growth rate, including no tradeoffs whatsoever (Fig. S8).

(iv) Non-sparse resource usage (such that every strain utilizes every resource, but at different rates), as well as extremely sparse resource usage (such that each strain is effectively a specialist, and the central limit theorem no longer applies).

We discussed these extensions in the Main Text (lines 105-6, 161-3, 321-3) as well as in Appendix 4 in the SI. Together, they help demonstrate that “out-of-the-box” applications of our analytical theory are often qualitatively (and even quantitatively) consistent with the numerical results, and the causes of any deviations can usually be explained in simple terms.

Overall, the study, albeit limited mostly to the scenarios with early mutations, is very thorough and detailed. At the same time, the presentation style leaves room for improvement. In the present form the manuscript does not appear reader-friendly to the general audience of Nature Communications.

We thank the Reviewer for this feedback. We have made various revisions to the manuscript to make it more reader-friendly for a broader audience. We describe these and other changes in more detail in our responses to the Reviewer’s specific suggestions below.

I would like to suggest:

(2) To specify resource dynamics in the main text

Thanks for the suggestion – we have revised Eq. 1 to explicitly specify the resource dynamics (which were previously only included in the SI).

(3) Eq. 1 needs to be first written in original, intuitively clear, terms such as $r_{\mu, i}$. Only after, if necessary for further analysis, it could be encoded in α_i 's and exponentiated logarithm. Apparently, the opening statement closely resembles that in Ref. (30), but that itself does not prove its suitability.

Thank you for this suggestion – we agree that the $r_{\mu, i}$ parameterization is more intuitive, at least up to the point that we start considering mutations. We have changed our notation to keep $r_{\mu, i}$ except when necessary. However, many of our key results turn out to be much simpler when expressed in $(\vec{\alpha}_\mu, X_\mu)$ basis, so we would prefer to keep this notation later on to enhance the readability of our work.

(4) Do upper case K_i in Figure 1 denote the same quantity as lower case k_i in the definition of h_i below Eq. 1.

We apologize for the confusion. To streamline notation, we now only use the original uppercase K_i variables in the Main Text, with the rescaled version κ_i only appearing in the SI (Eq. S12).

(5) What is R_0 ? It's hard to get what the symbol “~” means here.

Thanks for pointing this out. We have rewritten this paragraph to more clearly define the parameter R_0 (lines 102-3). In our original community assembly model, each strain has an independent probability R_0/R of using each resource, so R_0 denotes the average number of resources used by each strain.

(6) It is necessary to explain why γ is defined proportional to R_0 ? Once again, it would be easier on a reader to retain the original notations ($r_{\mu,i}$) where possible in the main text and use the derived ones for replica integration, etc.

Thanks for this suggestion. Upon further reflection, we agree that the γ notation was not necessary for the Main Text, and we have updated the manuscript to the magnitude of $\Delta\vec{\alpha}_\mu$ directly. This new notation makes it easier to see that the relevant scale of $\|\Delta\vec{\alpha}_\mu\|$ is proportional to the phenotypic variation among sampled species.

(This is an example where writing the same expressions in the original $r_{\mu,i}$ notation is much more cumbersome, illustrating the utility of the $\alpha_{\mu,i}$ basis for revealing the important features of the problem.)

(7) "For example, if a resource has a lower relative availability ($g_i < 0$), then it is not worth devoting energy to consume it, and a knock-out mutation for the resource should be beneficial." I don't think this energy is accounted for in the model.

Thank you for pointing this out. We were previously using "energy" in a colloquial sense to refer to the total uptake budget. However, we agree that this could be confusing, so we have changed this phrasing to be more consistent with our other descriptions of this concept.

(8) What is δk ?

We had previously used this notation to refer to the spread in resource supply rates. However, in an effort to reduce the mathematical notation in the Main Text, we have eliminated this variable and now refer to the resource supply rates \vec{K} directly.

(9) The term ϵ in Fig. 2 needs to be defined.

Thanks for this suggestion – we have removed all mention of ϵ outside of the SI, instead referring to its more direct meaning as the standard deviation of total uptake budget X_μ .

(10) "However, since the mean of the DFE is still centered at $s = 0$, the overall fraction of beneficial mutations remains constant as $S^ \rightarrow R$ (Fig. 2C)." I find this statement rather misleading since the probability of fixation, usually estimated as the fraction of invasion fitness and the birth rate, becomes negligible when $S^* \rightarrow R$.*

Thanks for pointing this out – we agree that the limit notation in this sentence was confusing, since it (incorrectly) implied that we were focused on the case where S^* becomes arbitrarily close to R . We have rewritten this sentence to use clearer non-mathematical language ("the overall fraction of beneficial mutations remains constant as niche saturation increases"), and to point out that a smaller number of them will survive genetic drift.

(11) Fig. S5 "Dashed line" -> blue line?

Thanks for catching this. We've corrected this in the revised manuscript.

(12) Once again, I suggest to move a large part of section 1.1 of Supplementary materials to the main text.

Thanks for this suggestion. We have moved the resource-explicit version of the model to the Main Text, as suggested above, and agree that this improves readability.

(13) To prove that all approximations are applicable, it would be illustrative to show that, for example, Fig. 2B is reproducible via direct integration of equations for population and resource dynamics. A self-consistency check that the consumption terms are always much larger than the dilution rate would also be convincing.

Thank you for this suggestion. We have implemented both of these ideas in the revised manuscript. First, we have expanded our derivation in SI Section 1.2, showing how the coarse-grained dynamics in Eq. 2 emerge from the resource-explicit dynamics in Eq. 1. This expanded derivation includes a new self-consistency condition that tells us when the consumption terms will dominate. Second, we have also complemented these analytical results by adding a new simulation figure (Figure S1), which confirms that direct integration of the resource-explicit dynamics yields predictions that closely match the direct optimization approach used in the rest of the paper. We note that this agreement continues to hold even when the resource supply terms are not extremely large compared to the dilution term (Fig. S1B, left), suggesting that the regime where dilution can be ignored is relatively wide.

(14) Eqs. 6-8 are unreadable in the main text. For example, I failed to figure out how for " $R_0 \gg 1$, the integral in Eq. (6) further reduces to eq. 8". I suggest to leave them in Supplementary materials and present the resulting Eq. (8), to illustrate the verbal explanations of various regimes.

Thank you for this feedback. We have completely rewritten this paragraph to eliminate some of the notation, and to write our results in a more easily understandable form. This new notation should better highlight the most important result, which is that coexistence occurs when the invasion fitness falls below a critical threshold (which is often much smaller than the size of a typical beneficial mutation). We have also added a new schematic figure in the main text (Fig. 3A) to demonstrate this principle visually.

(15) I could not see how (S4) is derived from (S1-S3)

Thanks for pointing this out. We have expanded the derivation of the resource dynamics in this section (and the Main Text, as suggested above), which should hopefully address this issue. This step comes from the fact that the total biomass approaches a constant at long times, which we now discuss explicitly in Section 1.2.

(16) Lines 707-708 This has been known long before (30,32), see, for example, (45)

Thank you for bringing this up. Strictly speaking, the MacArthur paper (Ref. 45) considers a slightly different model, so we wanted to make sure to include a reference that applies for the specific dynamics in Eq. 2. However, we agree that all are nearly equivalent for the relevant

conclusions in our paper, so we have added a citation to Ref. 45 in this sentence as well (and we have also changed the wording to emphasize that these are all well-known results).

Reviewer #2 Comments:

McEnany and Good study a resource-competition model with phenotypic mutations and describe some properties of random ecosystems invaded by such mutants. A key question that the authors ask is whether new mutants generally knock-out or coexist with their parent. They show that mutants can generally coexist with their parents, instead knocking out other metabolically distant species. The species most likely to go extinct are the ones at low abundance. Though the authors' calculations pertain only to the first steps of this eco-evolutionary process of mutants invading and modifying the community, the authors show numerical results suggesting that these general lessons tend to hold even after many steps of eco-evolution.

The paper is well-written and addresses a very interesting topic in the field. Eco-evolutionary dynamics have largely remained difficult to study from the ecological direction. Most population genetics assume that the rest of the community is implicit.

We thank the Reviewer for these positive comments about our manuscript.

However, our chief concern is that the authors derive their results in a very specific regime of the model where all species are effectively neutral (detailed below), which raises questions about the generality of the results. We believe the authors should address this assumption in the main text, and appropriately tone down their claims. Below, we detail our chief concern as well as highlight a few others, which we recommend the authors to address in a revised version of the manuscript.

We thank the Reviewer for raising this issue. However, as we describe in more detail below, we respectfully disagree with the suggestion that we have limited ourselves to a regime where all species are effectively neutral (which would be an important limitation if true). Rather, we would argue that non-neutral effects – encompassing both niche- and fitness-differences – are actually critical for many of our main results, including the stable coexistence of mutants with their parents, and the deterministic extinction of metabolically distant strains.

That being said, we agree with the Reviewer that this is an extremely important point, and that some of the confusion could stem from our current lack of clarity about the role of certain assumptions in our model. We have therefore revised the manuscript to clarify the assumptions that are required (see below), and how they do (or don't) relate to the concept of neutrality. We have also added a new analysis of a regime where our standard assumptions are strongly violated, and show that many of our qualitative conclusions continue to hold in this case as well. Together, we believe that these and other changes should address the Reviewer's chief concern about the generality of our results.

MAJOR COMMENTS

(1) The main assumption that allows the authors to calculate the key quantities of interest (e.g., \mathbb{P}_{coex}) is stated in section 1.2 of the SI. The authors decompose $\tilde{h}_i = 1 + g_i$ and Taylor expand $\tilde{F}(g_i)$ for small g_i . This decomposition (and retaining only $g_i + g_i^2$ terms) implicitly

assumes that rescaled resource availability for any resource is 1 near equilibrium, forcing the system to stay near a nearly neutral point where all resource availabilities are ≈ 1 . This effectively makes all species fitnesses $X_\mu \approx 1$, making them nearly neutral. This assumption is what allows the authors to frame their problem as an optimization problem, to which replica-theoretic techniques can be applied. This is a rather strong assumption that makes the analysis non-generic in our opinion, and that to our knowledge is not discussed in the main text. We suggest the authors explicitly mention this in the abstract, results, and/or discussion, since it is not clear if their results hold outside of this nearly neutral regime.

We appreciate this close reading of our theoretical results. First, a small clarification: our ability to map our problem to a convex optimization problem does *not* require any assumptions about the magnitudes of X_μ or g_i , but only that the strain dynamics can be described by the coarse-grained model in Eq. 2. In hindsight, we can see how our discussion of this point in the old version of SI Section 1.2 was potentially confusing, so we have rephrased this section (now SI Sections 1.5 & 1.6) to make this fact more clear. Crucially, all of our numerical simulations were performed by optimizing the full model in Eq. 2, providing an implicit check of the validity of all of our downstream analytical approximations.

However, we also appreciate the Reviewer's broader concern that our analytical solution of the optimization problem does place additional constraints on the magnitudes of g_i (and therefore X_μ). At the same time, we do not believe that "nearly neutral" is the appropriate label to describe the assumptions required for our analytic results to hold, for several reasons that we will discuss below.

As the Reviewer noted, the primary assumption required for our analytical calculation is that the excess resource availabilities, g_i , are small compared to one. Crucially, the g_i do not need to be *infinitesimally* small (as in the "nearly neutral" point in Ref. 32), but only small in comparison to this extrinsic scale. In practice, we find that g_i values as large as one (corresponding to resource availabilities that differ by up to ~ 2 -fold) are still reasonably well approximated by our analytical predictions (Fig. 3, S7B-C). Thus, the g_i condition is best viewed not as a statement about neutrality, but rather our ability to approximate the nonlinear dynamics in Eq. 2 by pairwise interactions between species. This latter feature is shared by many other ecological models, including the classic MacArthur model in Ref. 45.

When does the $g_i \lesssim 1$ approximation apply? Eq. 4 shows that this assumption will hold as long as the spread in the resource supply rates is not too large ($|K_i - \bar{K}| \lesssim \bar{K}$), and the number of surviving species is not too small ($S^* \gtrsim R/R_0$). This latter condition has a simple interpretation as the requirement that each resource must be consumed by multiple surviving strains. This is a natural requirement for large-N approaches like replica theory to apply (and is shared by related works like Refs. 29 and 36).

How does this relate to X_μ ? Eq. S118 shows that when our g_i approximation is valid, the spread in the total uptake budgets of the S^* *surviving species* must also be less than one. Note that this is true even when the variation in the initial species pool is much larger (as is the case for some of the examples in Fig. 3B). In other words, the community assembly process dynamically selects for an initial community where the total uptake budgets lie in a narrower range. Crucially, this result is a *consequence* of our community assembly model in the limit of a large community with a modest amount of metabolic overlap, rather than an additional assumption about the allowed values of X_μ . Despite the community selecting for a narrower

range of X_μ , the differences in total uptake budget between species can drive significant differences in the steady-state communities we study.

We note that growth rate differences on the order of 0.01-0.5 are quite large by evolutionary standards, and are consistent with recent empirical observations in *in vitro* communities of soil bacteria (Ref. 61). This suggests that the $g_i \lesssim 1$ approximation is an empirically relevant regime, rather than a corner case designed to make the math work.

Changes in the revised manuscript. We appreciate that these are subtle points, so we have made various changes to the revised manuscript to clarify this issue. First, we have added some additional discussion of the conditions of validity for our replica-theoretic calculations in the Main Text, and in a new dedicated section in the SI (3.5, “Self-Consistency of Replica Assumptions”). We have also included a new Supplementary Figure (Fig. S7) demonstrating that our approximations continue to hold when we allow for larger variations in the resource supply rate ($|K_i - \bar{K}| \sim \bar{K}$). Finally, to consider what happens under sampling schemes that allow for much wider fitness differences ($|X_\mu - \bar{X}| \gg 1$), we have added a new analysis of a “specialist assembly” process, where each resource is consumed by a single species (SI Section 4.5; Fig S3D). This minimal niche overlap allows for high levels of niche saturation, even when the surviving species have very different X_μ values. We find that although these communities are far from the regime described by our replica-theoretic calculations, many of our qualitative results continue to hold, such as the observation that mutants can coexist with their parents in saturated communities, while driving metabolically distant species to extinction.

We hope that these changes better illustrate that our results do not rely on any explicit neutrality assumptions (as suggested in the Reviewer’s original comment above) but are instead a more generic consequence of starting from a large, randomly assembled community. We have made various small changes to the abstract and main text to better emphasize this latter assumption.

(2) Though the presentation is clear in most of the places, there are few places where it could be clearer. For example:

The physical meaning of D (in S57) could be explained in more words. In section 3.3 (scaling analysis of \mathbb{P}_{coex}), the authors state that $D > E$ represents the requirement that the mutant is well-adapted enough to survive in the population. We could see it somewhat, but it could be made more apparent.

Thanks for this suggestion. In the revised manuscript, we have added a new comment about how D and E relate to the resource deficit experienced by the parent and mutant strains when these terms are first mentioned, and we have reworked SI Section 3.3 to more clearly explain how the “noise terms” in D and E relate to physical properties of the community.

(3) We can somewhat see that S87 represents the relative abundance of parent species (as $\frac{2}{S'(\lambda)} \approx \frac{1}{S^}$ in large $|\lambda|$ limit) but not exactly.*

Thanks for bringing this to our attention. We have revised SI Section 3.3 to clarify how the relative abundances are related to the noise variable Z_2 . As part of this, we have redefined $Z_2 - \lambda$, which more intuitively relates Z_2 to a positive “fitness” variable: if $Z_2 > \lambda$, the species will survive, and $Z_2 - \lambda$ is proportional to its abundance.

MINOR COMMENTS

(4) Since $\sum_{\mu=1}^P n_{\mu}(t) = \sum_{i=1}^R \kappa_i + \frac{\sum_{\mu=1}^P n_{\mu}(t) - \sum_{i=1}^R \kappa_i}{e^{-t}}$. Dividing by $\sum_{i=1}^R \kappa_i$ and $\sum_{\mu=1}^P n_{\mu}$ are not the same thing in equation S2.

Thanks for pointing this out. We have expanded this section of the SI in response to Comments #2 and #15 from Reviewer 1. Our rewrite of Section 1.1 in the SI should account for this more clearly, since it explicitly discusses the total biomass of the community. Since we are primarily interested in the introduction of a mutant at low frequency to an already-stable community (as opposed to the initial approach of that community toward equilibrium), we do not expect total biomass to change, enabling us to make this approximation.

(5) Equations S6a and S6b don't seem quite right if in S6b, the second term in the brackets is meant to represent the unutilized resources. Because if l_i is the fraction of i^{th} resource that is utilized by some species, then it is $(1-l_i)$ that is excreted back to the environment.

Thank you for pointing out this error; we have corrected this block of equations in the revised manuscript.

(6) In section 2.5 (Parent-Mutant Coexistence Probability), authors have only considered the global minimum for $e^{-\frac{1}{2} \vec{v} \mathbf{M}^{-1} \vec{v}}$, whereas there exist cases in which global minimum of this function, can lie outside the non-positive-resource-surplus region and still have a point in the constrained region for which the function is optimum.

We appreciate your close reading of our derivation, and have rewritten this section to make each step more clear. In this case, we note that the positive definite form of the matrix M causes this minimization problem to reduce to a convex optimization problem. This implies that the minimum over the constrained region must either correspond to the global minimum, or a minimum on one of the boundaries. This latter case is considered in Eqs. S79-S85.

REVIEWERS' COMMENTS

Reviewer #1 (Remarks to the Author):

I think the Authors have successfully addressed all comments and suggestions made in the previous review and the the manuscript can be accepted for publication. Perhaps a couple of minor points:

Why not to include the definition of $f_{\{\mu\}}$ (S6) in the main text?

The sentence in lines 336-338, "Previous work has suggested that even distantly related strains could exhibit correlated dynamics if their resource consumption strategies are anomalously similar to each other (26)." seems to have an internal contradiction: In the present context the only feature of a strain is its set of resource strategies. So if resources strategies are similar, the strains are related and vice versa.

Reviewer #2 (Remarks to the Author):

I thank the authors for engaging with and responding to all my comments. I am satisfied with the revised manuscript and recommend it for publication.

Detailed Response to Reviewer Comments

Reviewer #1 Comments:

I think the Authors have successfully addressed all comments and suggestions made in the previous review and the the manuscript can be accepted for publication. Perhaps a couple of minor points:

Why not to include the definition of f_{μ} (S6) In the main text?

We thank the reviewer for their recommendation of our paper, and have incorporated this suggestion as to f_{μ} .

The sentence in lines 336-338, "Previous work has suggested that even distantly related strains could exhibit correlated dynamics if their resource consumption strategies are anomalously similar to each other (26)." Seems to have an internal contradiction: In the present context the only feature of a strain is its set of resource strategies. So if resources strategies are similar, the strains are related and vice versa.

Thank you for pointing this out; we have made this point more clear in the manuscript. In particular, the “phylogenetically distant” strains we discuss are those identified in Ref. 26 (where actual phylogenetic distance was used). While our model does not include phylogenetic distance as an explicit parameter separate from metabolic distance, we can consider a somewhat analogous hypothesis: that some pairs of independently sampled (“unrelated”) strains might have closer metabolic similarity than usual, and that mutation in one of these strains might be correlated with extinction with its pair. We have clarified the distinction between these concepts in the text.

Reviewer #2 Comments:

I thank the authors for engaging with and responding to all my comments. I am satisfied with the revised manuscript and recommend it for publication.

Thank you for the positive recommendation; we are excited for publication in Nature Communications!